# Early Detection of Red Palm Weevil, *Rhynchophorus ferrugineus* (Olivier), Infestation Using Data Mining

**DOI:** 10.3390/plants10010095

**Published:** 2021-01-06

**Authors:** Heba Kurdi, Amal Al-Aldawsari, Isra Al-Turaiki, Abdulrahman S. Aldawood

**Affiliations:** 1Computer Science Department, College of Computer and Information Sciences, King Saud University, Riyadh 11451, Saudi Arabia; 437203624@student.ksu.edu.sa; 2Mechanical Engineering Department, Massachusetts Institute of Technology (MIT), Cambridge, MA 02142-1308, USA; 3Information Technology Department, College of Computer and Information Sciences, King Saud University, Riyadh 11451, Saudi Arabia; ialturaiki@ksu.edu.sa; 4Plant Protection Department, College of Food and Agriculture Sciences, King Saud University, Riyadh 11451, Saudi Arabia; aldawood@ksu.edu.sa

**Keywords:** red palm weevil, *Rhynchophorus ferrugineus*, palm, infestation, prediction, data mining

## Abstract

In the past 30 years, the red palm weevil (RPW), *Rhynchophorus ferrugineus* (Olivier), a pest that is highly destructive to all types of palms, has rapidly spread worldwide. However, detecting infestation with the RPW is highly challenging because symptoms are not visible until the death of the palm tree is inevitable. In addition, the use of automated RPW weevil identification tools to predict infestation is complicated by a lack of RPW datasets. In this study, we assessed the capability of 10 state-of-the-art data mining classification algorithms, Naive Bayes (NB), KSTAR, AdaBoost, bagging, PART, J48 Decision tree, multilayer perceptron (MLP), support vector machine (SVM), random forest, and logistic regression, to use plant-size and temperature measurements collected from individual trees to predict RPW infestation in its early stages before significant damage is caused to the tree. The performance of the classification algorithms was evaluated in terms of accuracy, precision, recall, and F-measure using a real RPW dataset. The experimental results showed that infestations with RPW can be predicted with an accuracy up to 93%, precision above 87%, recall equals 100%, and F-measure greater than 93% using data mining. Additionally, we found that temperature and circumference are the most important features for predicting RPW infestation. However, we strongly call for collecting and aggregating more RPW datasets to run more experiments to validate these results and provide more conclusive findings.

## 1. Introduction

The red palm weevil (RPW), *Rhynchophorus ferrugineus* (Olivier), is a highly destructive pest that affects palm species worldwide [1]. The RPW threatens many types of palm trees, including coconut, sugar, oil, sago, palmyra, royal, Washingtonian, and date palms [2]. Infestation with the RPW was first detected in the mid-1980s in the Arabian Gulf region. Since then, it has spread rapidly worldwide, reaching the Middle East, Southern Asia, North Africa, Russia, Spain, and many other regions [2,3]. It has been reported that up to 30% of date palm production can be lost due to pests and disease [4]; millions of dollars are spent on pest control and the removal of infested trees. Recently, infested palms have been burned to minimize the spread of the RPW and save other palms from infestation.

Early detection of RPW infestation is vital for a more effective control of the pest damage. However, the entire life cycle of RPW larva is concealed inside the palm, which hinders the detection of an infestation in its early stages [5,6]. Current early detection methods include visual inspection of the tree to observe symptoms, detection of the sound produced by feeding larvae, chemical detection of volatile signatures produced by infested date palms, and thermal imaging to detect temperature increases in infested palms [7]. The classical methods might be in some cases more expensive (e.g., sound detection); however, the observation of symptoms is the most reliable method to detect RPW. However, as symptoms only show in late stages, the detection and prediction of RPW infestations in early stages remains a significant challenge. Therefore, many researchers from different fields have been motivated to work in this area. Rather than relying on traditional detection methods, there is a need to develop computational methods that are reliable and sustainable for detecting and managing RPW populations to reduce the potential losses of palms. It is also necessary to identify the key features that are relevant to the identification of infested palm trees.

Data mining is the process of identifying hidden patterns and relationships in large amounts of data [8]. The potential of data mining techniques has been demonstrated in many fields, such as healthcare, business, and education. Data mining techniques have also been applied in agricultural problems, with results exhibiting high potential [9,10,11]. However, to our knowledge, none of the previous works studied RPW; this can be attributed to the unavailability of published datasets. In this paper, we assessed the capability of ten state-of-the-art classification algorithms, Naive Bayes (NB) [8], KSTAR [12], AdaBoost [13], bagging [14], PART [15], J48 decision tree [16], multilayer perceptron (MLP) [17], SVM [18], random forest [19], and logistic regression [20], which was built based on real data collected from Saudi Arabia [21] as part of a large national project with a goal of aiding in reducing the loss of palm trees by enabling the administration of available treatment and injection before losses occur. The rest of this paper is organized as follows: related literature is reviewed in Section 2. Section 3 describes the materials and methods used, while Section 4 presents and discusses the experimental results. Finally, the conclusions of the paper are stated in Section 5.

## 2. Literature Review

Data mining techniques have been applied successfully in a wide range of fields, such as healthcare [22], business [23], and education [24]. In agriculture-related fields, Ref. [25] presented one of the earliest surveys on the use of data mining techniques to address agricultural problems. Data mining has exhibited potential when applied to the prediction of wine fermentation, estimation of soil water parameters, recognition of animal sounds, and detection of meat and bone meal. The authors highlighted the use of four data mining techniques: artificial neural networks (ANNs), SVMs, k-nearest neighbor (KNN), and k-means.

For pest disease prediction, [26] presented a system that utilized data mining techniques and wireless sensor networks. The goal was to determine the correlations among weather, pests, and diseases. The dataset used in their study represented real-time data collected using GPRS through four seasons. The classification and correlation analysis were performed using the Gaussian Naive Bayes algorithm. The results revealed that the cumulative model was more accurate than the empirical model because of its proximity to the ground level data. In particular, it was effective in saving crops from pests and diseases and improving crop yields.

Researchers in [27] used decision-tree algorithms to predict crop productivity. For the study, a crop dataset was obtained from the Ethiopian Economic Association. Three decision tree algorithms were used: J48, REPTree, and random forest. The authors concluded that the attribute “fertilizer use” exhibits the highest predictive power. In addition, REPTree outperformed the other classifiers and was followed by random forest.

Data mining was also used to assist crop protection decisions on kiwifruit in New Zealand. Researchers in [28] presented a model that used the data mining algorithms decision tree, Naive Bayes, random forest, AdaBoost, SVM, and logistic regression. The classifiers were applied on spray diary and pest monitoring datasets. The performance of these models was evaluated in terms of precision and recall measures. The results revealed that the models with a few attributes provided effective predictions. Moreover, AdaBoost outperformed the other classifiers and was followed by the Naive Bayes classifier.

In [29], a model for rice disease prediction was developed. The proposed incremental model was based on the particle swarm optimization (PSO) technique and association-rule mining. It was designed to determine an optimal number of classification rules. The classification accuracy was evaluated on a test dataset and compared with that of other state-of-the-art classification methods, such as Naive Bayes, KSTAR, AdaBoost, bagging, PART, J48, MLP, SVM, random forest, logistic regression, and PSO. The incremental model obtained the highest accuracy in three out of the five datasets. The proposed method reduced the computational time and demonstrated its significance and effectiveness.

For early detection of cherry fruit pathogen diseases, [30] proposed a system that used four data-mining techniques: a linear discriminant model, quadratic discriminant model, pseudo-linear discriminant model, and compact classification tree (CCT) model. The accuracy of the models was measured, and a comparison between the models was presented to select the best model. The test dataset was created randomly in ten iterations from the training dataset, and the models were tested on these ten iterations. The evaluation results revealed that the CCT model exhibited 93.6% accuracy, which was the highest among all the models in all 10 testing iterations.

SVMs with different kernels and acoustic signal extraction were used to detect termite infestation [31]. The experimental results revealed that the SVM with the polynomial kernel function achieved high classification accuracy.

Recently, researchers in [32] used data mining to address the problem of pepper *Fusarium* disease detection. Light reflections from the pepper leaves, measured using a spectroradiometer, were employed to categorize the plant into four classes: healthy, fusarium-diseased mycorrhizal fungus, and fusarium-diseased and mycorrhizal fungus. Experiments were conducted using ANNs, Naive Bayes, and KNN for classification. The results with high accuracy values demonstrated the effectiveness of data mining techniques.

A wireless visual sensor network was implemented to monitor paddy crops for weeds [33]. Images were collected and processed to remove the soil background, and different shape features were extracted. The system implements random forest and SVM classification algorithms. The results and observations obtained from the experimental setup of the system revealed that the random forest classifier outperformed the SVM classifier.

Notwithstanding the previous efforts, there is still inadequate research into the detection and prediction of RPW infestation using data mining techniques. We consider this to be due to the challenge of obtaining representative datasets of infestation. In [34], an automated system based on ANNs was developed for identifying RPW to aid RPW detection and prediction. The proposed ANN model was trained and evaluated using an image dataset containing 326 RPW images and 93 images of other insects. The authors concluded that a three-layer ANN using a conjugate gradient with the Powell-Beale restarts algorithm for feed-forward supervised learning is optimal for identifying red palm weevils. However, the proposed system was computationally demanding. To overcome this limitation, Ref. [35] used a smaller number of image descriptors in combination with ANN models. The new system provided better identification of RPWs and was observed to be up to 14 times faster in training and three times faster in testing insect images.

However, the two previous RPW studies are based on an image dataset, which means that the infested palm trees were already at a severe infestation level at which the symptoms are visible. In contrast, our study aims at early detection and prediction before symptoms become visible, so it does not rely on images, which also makes implementation of the system more feasible when storage and computation overheads are considered.

## 3. Materials and Methods

In this research, we evaluated the performance of ten classification algorithms in predicting palm tree infestation with the RPW. All of the classification algorithms were implemented using the Waikato Environment for Knowledge Analysis (WEKA) and run on a MacBook Pro 9.2 with an OS X 10.9.5 operating system and a 2.9-GHz Intel Core 17 with 8 GB memory. All the models are evaluated using 10-fold cross-validation. The dataset used in this paper is a real dataset of palm trees in Saudi Arabia, collected after cutting the trees to detect infestation and applying a treatment [21].

### 3.1. Dataset

The dataset represents an experiment that was conducted on date palm trees in the Kharj area of Riyadh, Saudi Arabia, in April 2017, for the purpose of testing a control method using nematode treatment (Koppert’s “Palmanem” (entomopathogenic nematode, Steinernema carpocapsae)) (Koppert Biological Systems, Rodenrijs, The Netherlands).

In this experiment, date palms were selected for being either infested or healthy. All tree attributes (15 attributes) where measured. Afterwards, the trees were dissected, then the presence/absence of infestation status was verified. The RPW dataset has 36 features and 1200 records. The dataset has some missing values that were replaced with the mode of each class. The dataset is imbalanced: 384 records belong to infested trees, whereas 816 records belong to healthy trees. We resampled the dataset to obtain an equal number of records, 600, for each class. Attributes with one value for all the records were removed. The final dataset consists of 15 attributes, as shown in Table 1: treatment, injection pattern, exposure time, replication, treatment date, plant height-1, plant height-2, diameter, circumference, north temperature, east temperature, south temperature, west temperature, coordinate (north), and coordinate (east).

### 3.2. Feature Selection

To determine the key features that contribute to the classification of a palm tree as infested or healthy, we computed Pearson’s correlation coefficient between the attributes and class. In addition, we computed the information gain and gain ratio of each attribute with respect to the classes [36]. In the RPW dataset, the attributes with a value of zero for any of the computed measures were removed.

### 3.3. Classification Algorithms

In this study, we considered 10 state-of-the-art classification algorithms, Naive Bayes (NB) [8], KSTAR [12], AdaBoost [13)], bagging [14)], PART [15)], J48 decision tree [16], multilayer perceptron (MLP) [17], SVM [18], random forest [19], and logistic regression [20], due to their popularity and availability in many data mining toolkits. Figure 1 shows the steps followed in this study.

### 3.4. Performance Measures

We evaluated the prediction performance of the selected classification algorithms in terms of the following measures:Accuracy: the percentage of correctly classified samples, calculated based on (1):
Accuracy = (TP + T N)/(TP + TN + FP + FN).(1)Recall: the fraction of samples correctly classified as infested out of the total number of infested samples, calculated based on (2):
Recall = (TP)/(TP + FN).(2)Precision: the fraction of samples correctly classified as infested out of all the samples predicted to be infested, calculated based on (3):
Precision = (TP)/(TP + FP).(3)F-Measure: the harmonic mean of precision and recall, calculated based on (4):
(4)F-Measure=(2 × precision × recall)/(precision + recall)
where TP (true positives) is the total number of samples that are correctly classified as infested, FP (false positives) represents the total number of samples that are incorrectly classified as infested, FN (false negatives) represents the total number of samples that are incorrectly classified as healthy, and TN (true negatives) represents the total number of samples that are correctly classified as healthy.

## 4. Results and Discussion

The values obtained for Pearson’s correlation coefficient revealed that the west, east, north, and south temperature attributes are highly correlated to the class label. The correlation values obtained are 0.63937, 0.51306, 0.49284, and 0.25765, respectively. Plant height-1 achieved a very low correlation value of 0.00316, indicating that it may be less indicative of an infestation. The overall correlation values fall within the range from 0.00316 to 0.63937.

The temperature attributes exhibit high information gain. The values are 0.71068, 0.71068, and 0.709856 for the north, east, and west attributes, respectively. The circumference also appeared to be important for classification, as it exhibits an information gain of 0.707054. Among all the attributes, exposure time, which describes the duration (in months) of exposure to treatment, exhibited zero information gain; therefore, it was removed. With regard to the gain ratio, the overall values were somewhat lower than those of the two previous measures. They fell within a range between zero and 0.401156. The highest-ranking attributes are related to the temperature and circumference. Furthermore, exposure time again appears to be irrelevant to the classification, as it again obtained the value zero, thus, it was removed. Therefore, we included all 15 attributes for the correlation attribute evaluation method, and we removed the exposure time attribute for the other two methods. Hence, we have 14 attributes for the information gain and gain ratio attribute evaluation methods.

Our findings in regard to the importance of temperature in predicting infestations with RPW are consistent with previous studies [37,38], showing that the internal temperature of the date palm trunk is an important attribute, as RPW infestation causes a significant change in the internal temperature of the date palm trunk compared to the environment. Accordingly, they suggested that the change in temperature can be used to predict the presence of RPW inside the date palm trunk. In addition to temperature, some researchers also reported that the RPW population inside the date palm trunk is positively correlated with the thickness (diameter/circumference) of the trunk [39]. The performance of the state-of-the-art algorithms on the balanced RPW dataset was very similar for all of the three feature selection methods, as illustrated in Table 2. All of the algorithms (except Naive Bayes) obtained highly similar results for the four performance measures: accuracy (93.08%), precision (87.80%), recall (100%), and f-measure (93.50%). Naive Bayes obtained lower values than those of the other algorithms in terms of accuracy (82.58%), recall (72.80%), and f-measure (80.70%), which can be attributed to the fact that it implicitly assumes that all attributes are mutually independent, which is unrealistic in real datasets.

## 5. Conclusions

Over the last few decades, palm trees have been increasingly infested by the RPW, a destructive pest. To minimize the loss of palm trees and control this pest, we need to detect infestation without needing to cut the trees. Therefore, the main objective of this study is to evaluate the capability of different data mining classification algorithms to accurately predict RPW infestation without losing palm trees. We applied 10 state-of-the-art classification techniques that have demonstrated effectiveness in different areas. These algorithms were run using 10-fold cross-validation on a real RPW dataset. We compared the ability of these classification algorithms to predict RPW infestation. The overall performance results of the classification algorithms were identical at 93% in term of prediction accuracy with Naïve Bayes showing the lowest accuracy at 82%. This is contrast to [29], where the same 10 classification algorithms were applied for rice disease prediction and Naïve Bayes showed the highest accuracy, 88%, while AdaBoosr had the low accuracy at 37% and the other classifiers varied slightly in the range (81–87).

With regard to the key features for classifying RPW, the experimental results indicate that circumference and temperature are important features for predicting infestation. The finding is aligned with previous studies [37,38] where they have found that RPW infestations cause a significant increase in the internal temperature of date palm trunk in comparison with healthy date palm and ambient atmospheric temperature. Additionally, in [39], it has been shown that RPW population inside date palm trunk revealed a positive correlation with the date palm trunk circumference. Nevertheless, the attribute ranking for the feature selection measures reveals the challenge in determining specific characteristics of RPW infestation.

However, the dataset used requires expansion and improvement for more conclusive findings. We strongly call for collecting and aggregating more RPW datasets to run more experiments to validate these results. In addition, the accuracy of RPW prediction may be improved by tapping the potential of unexplored data mining techniques.

## Figures and Tables

**Figure 1 plants-10-00095-f001:**
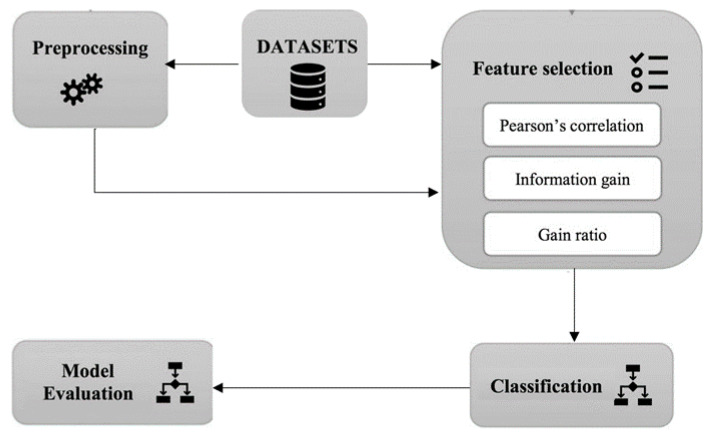
The workflow of this study.

**Table 1 plants-10-00095-t001:** Descriptions of the dataset attributes.

Attribute	Definition
Treatment	In the experiment, each treatment represents a control tactic (with an insect nematode (Koppert’s “Palmanem” (entomopathogenic nematode) or water tactic)
Injection pattern	Tactics are administered into the date palm through injections. Injection pattern reveals the position of injection into the date palm i.e., injection at the base of the tree, injection at one meter height, injections in a spiral manner to cover the entire trunk (north, south, east and west).
Exposure time	It is a span of time for which a treatment is applied to the experimental units.
Replication	Repetition of the experimental treatments.
Treatment date	Date of treatment application.
Plant height-1	Length of the date palm tree from base to the area below the crown area.
Plant height-2	Total length of the date palm tree.
Diameter	The total width of the date palm trunk.
Circumference	The total distance around the palm trunk.
North temperature	Internal temperature of the date palm trunk in the north direction.
East temperature	Internal temperature of the date palm trunk in the east direction.
South temperature	Internal temperature of the date palm trunk in the south direction.
West temperature	Internal temperature of the date palm trunk in the west direction.
Coordinate (north)	Coordinate associated with positions in the north direction.
Coordinate (east)	Coordinates associated with positions in the east direction.

**Table 2 plants-10-00095-t002:** Performance of the 10 classifiers on the red palm weevil (RPW) dataset.

Algorithm	Accuracy	Precision	Recall	F-Measure
Naive Bayes	82.58%	90.50%	72.80%	80.70%
Logistic	93.08%	87.80%	100%	93.50%
MLP	93.08%	87.80%	100%	93.50%
SVM	93.08%	87.80%	100%	93.50%
KSTAR	93.08%	87.80%	100%	93.50%
AdaBoost	93.08%	87.80%	100%	93.50%
Bagging	93.08%	87.80%	100%	93.50%
PART	93.08%	87.80%	100%	93.50%
J48	93.08%	87.80%	100%	93.50%
Random Forest	93.08%	87.80%	100%	93.50%

## Data Availability

Not applicable.

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
