# Peer review of "Early Detection of Red Palm Weevil, *Rhynchophorus ferrugineus* (Olivier), Infestation Using Data Mining"

_plants, 2021, doi:10.3390/plants10010095_

Round 1
Reviewer 1 Report
The manuscript of Kurdi and colleagues presents a very interesting aspect of early detection of one of the more infamous insects, the Red Palm Weevil. The manuscript is in general well written and there are only some minor points that need to be addressed. The main point that in my opinion needs particular attention, is the structure of the manuscript, as in some cases the authors provide information that should be removed to other parts (see below for these points in details). Finally, the authors should enhance conclusions with references of previous studies, as this will vividly show the importance and contribution of their approach.
Line 42:...for a more effective control...
Line 45: ..of signature lures...
Line 47: The classical methods might be in some cases more expensive (f.e. sound detection) but, at least the observation of symptoms, is the most reliable method to detect RPM. Please rewrite.
Lines 65-70. This paragraph includes your results briefly. Please remove from here, and maybe integrate them in the respective (Results) part.
Line 137: ...computationally demanding...
Lines 174-187: The information you present here belongs to Materials and Methods. Please move there.
Lines 217-228: the part of conclusions needs to be enhanced by providing a comparative approach with other previous studies on RPW or even other insects as well. For that, the authors can definetely use the studies they presented in the Literature Review section. To the same direction, a comparison with the performance of other algorithms employed with RPM (in a first approach) or even other insects (in a second approach) could be very interesting.
Reviewer 2 Report
This is a useful study about use of data mining techniques to estimate the presence of red palm weevil infestations from measurements of the size and temperature attributes of individual trees. The manuscript might benefit from a brief statement about the values of attributes that were most indicative of RPW infestation if that is possible to summarize.
Otherwise, the reviewer has only a few suggestions to make at lines listed below:
Line 18 “In addition, the use of automated RPW weevil identification tools to predict infestation is complicated by a lack of RPW”
Line 21 “to use plant-size and temperature measurements collected from individual trees to predict RPW infestation in its early stages before significant damage is caused to the tree.”
Line 42 “better control of the pest damage.”
Line 158 The attribute characteristics should be explained more clearly if possible. For example what is exposure time and treatment? Exposure time was not defined until line 200.
Line 187 “(Precision+recall)
